# Translucent and Highly Toughened Zirconia Suitable for Dental Restorations

Seiji Ban [1,*] , Yuta Yasuoka [2], Tsutomu Sugiyama [2] and Yuzo Matsuura [2]

1   Department of Dental Materials Science, School of Dentistry, Aichi Gakuin University, Nagoya 464-8650, Japan
2   KCM Corporation, Nagoya 455-8668, Japan
*   Correspondence: sban@g.agu.ac.jp; Tel.: +81-052-751-2561

**Abstract:** Background: There is a limit to improving the characteristics of zirconia with only one kind of stabilizing element such as yttrium. The purpose of this study is to systematically evaluate the effects of various co-doped elements on mechanical and optical properties and to develop a novel composition of zirconia with enough properties to apply to dental restorations. Methods: Forty-four kinds of zirconia were prepared by combining trivalent cations yttrium (Y) and ytterbium (Yb), and pentavalent cations niobium (Nb) and tantalum (Ta) oxide as stabilizers. The combined contents ranged from 0 to 5.6 mol% for $Y_2O_3$, 0 to 4.2 mol% for $Yb_2O_3$, 0 to 1.5 mol% for $Nb_2O_5$, and 0 and 1.2 mol% for $Ta_2O_5$. These specimens were determined for fracture toughness and opacity. X-ray diffraction studies were undertaken to evaluate the microstructural change. Results: The present study revealed that adding of the trivalent cations Y and Yb reduced fracture toughness and opacity, whereas the addition of pentavalent cations Nb and Ta to zirconia stabilized with trivalent cations increased both properties. There was no clear difference in the effects of Y and Yb, Nb, and Ta. Conclusions: Considering many factors, the following composition is optimal: 3–4.2 mol% $Y_2O_3$ and/or $Yb_2O_3$ stabilized zirconia with up to 1.5 mol% $Nb_2O_5$ has sufficiently high fracture toughness values and sufficiently high translucency suitable for dental restorations.

**Keywords:** fracture toughness; opacity; translucency; zirconia; co-doping; dental applications





## 1. Introduction

Zirconia, in which a small amount of rare earth oxide such as yttrium oxide ($Y_2O_3$) is contained as a stabilizer, has high strength and high toughness. When yttria is 3 to 8 mol%, tetragonal and cubic phases are mixed at room temperature, and it is called partially stabilized zirconia (PSZ). When yttria is around 3 mol%, the tetragonal phases are close to 100% at room temperature, and it is called tetragonal zirconia polycrystal (TZP). In the early stages of zirconia application in dentistry, this yttria 3 mol% tetragonal zirconia polycrystal (3Y-TZP) was introduced in dentistry as a core material instead of metal. When 3Y-TZP is used as dental prostheses, the surface must be veneered with feldspathic porcelain to improve its appearance due to low translucency. To take full advantage of the strength of zirconia, it should not be veneered with porcelain. Therefore, high translucent PSZ has been developed to produce monolithic zirconia prostheses [1]. Although PSZ has a more complicated structure than TZP and the manufacturing process should be more strictly controlled, it is possible to provide stable materials by many manufacturers.

To increase the translucency of PSZ, it is conceivable to increase the content of a stabilizer such as yttrium oxide [2,3]. Further, since PSZ is only partially stabilized, there is the problem that long-term stability is particularly difficult in a hydrothermal environment [4]. For example, a PSZ undergoes a phase transition from tetragonal to monoclinic when heated in the presence of moisture. There is an inherent problem that the strength of the sintered body decreases due to the growth of microcracks caused by the volume expansion of about 4% accompanying this phase transition. Therefore, to utilize PSZ in an industrial

product, it is necessary to use a sintered body in which the progress of the phase transition is sufficiently suppressed according to the application of the industrial production in the environment in which it is used.

To suppress the progress of the phase transition of PSZ in a hydrothermal environment, it is conceivable to increase the content of a stabilizer such as yttrium oxide. However, if the content of the stabilizer is increased, the mechanical properties such as flexural strength and toughness deteriorate, and if the content of the stabilizer is reduced, the deterioration of the mechanical properties can be avoided [3]. However, there is a risk that the progress of the phase transition in the sintered body cannot be sufficiently suppressed [5,6]. PSZ using cerium (Ce) as a stabilizer improves its stability and fracture toughness when the content of cerium is increased [7–9]. It has not been used for dentistry because its color is yellowish, and it is greatly affected by the firing atmosphere. Therefore, there is a limit to improving the characteristics of zirconia with only one kind of stabilizing element.

Under such circumstances, various PSZs have been conventionally proposed. For example, increasing alumina ($Al_2O_3$) achieved enhanced strength and aging resistance but with opacities precluding use as anesthetic dental restorations [10]. In addition, PSZ in which two or more stabilizing elements coexist have been studied: e.g., yttrium-ytterbium (Y-Yb) [11], yttrium-niobium (Y-Nb) [12–15], yttrium-cerium (Y-Ce) [16], yttrium-tantalum (Y-Ta) [17–19], yttrium-niobium/tantalum (Y-Nb/Ta) [20], niobium-tantalum (Nb-Ta) [21], and yttrium-tantalum-niobium-hafnium (Y-Ta-Nb-Hf) [22]. These co-doping methods have high possibilities for developing novel zirconia. Although various constitutions of PSZ sintered bodies have been proposed as mentioned above, no compositions have both high strength and high translucency. It is desired to develop novel zirconia having both properties. However, the effects of co-doping on these properties are complex and no practical composition has been determined.

The purpose of this study is to systematically evaluate the effects of these elements co-doped as stabilizers on mechanical and optical properties and to develop a novel composition of PSZ with enough properties suitable for dental restorations.

## 2. Materials and Methods

### 2.1. Materials

Forty-four kinds of zirconia were prepared with stabilizers obtained by combining yttrium, niobium, ytterbium, and tantalum oxide as listed in Table 1. After the mixing of $ZrO_2$ sol, $YCl_3$, and/or $YbCl_3$, the pastes were neutralized, dehydrated, dried, and pulverized into powders. The powders were calcined to produce a mass of solid solution. The mass was crushed and pulverized to fine powders in a pot mill. After aggregation by neutralization, the mixtures were dried again and crushed in a mortar. Then, powders *A* (No. 1–9) without Nb nor Ta were completed. $Nb_2O_5$ and/or $Ta_2O_5$ powders were added to powder *A* and mixed for 1 h in a pot mill. After aggregation by neutralization, the mixtures were dried again and crushed in a mortar. Then, powders *B* (No. 10–44) with Nb and/or Ta were completed.

**Table 1.** Chemical composition of stabilizers and their fracture toughness and opacity.

| Group | No | Content (mol%) | | | | Toughness (MPa$\sqrt{m}$) | Opacity (%) |
|---|---|---|---|---|---|---|---|
| | | $Y_2O_3$ | $Yb_2O_3$ | $Nb_2O_5$ | $Ta_2O_5$ | | |
| $Y_2O_3$ | 1 | 2.0 | 0.0 | 0.00 | 0.00 | 9.1 | 86.4 |
| | 2 | 3.0 | 0.0 | 0.00 | 0.00 | 4.4 | 81.0 |
| | 3 | 4.2 | 0.0 | 0.00 | 0.00 | 3.6 | 72.9 |
| | 4 | 5.6 | 0.0 | 0.00 | 0.00 | 3.0 | 70.0 |
| $Yb_2O_3$ | 5 | 0.0 | 4.2 | 0.00 | 0.00 | 3.7 | 73.2 |

**Table 1.** *Cont.*

| Group | No | Content (mol%) | | | | Toughness (MPa√m) | Opacity (%) |
|---|---|---|---|---|---|---|---|
| | | Y$_2$O$_3$ | Yb$_2$O$_3$ | Nb$_2$O$_5$ | Ta$_2$O$_5$ | | |
| Y$_2$O$_3$-Yb$_2$O$_3$ | 6 | 1.8 | 1.7 | 0.00 | 0.00 | 4.2 | 73.8 |
| | 7 | 1.8 | 2.4 | 0.00 | 0.00 | 3.8 | 72.4 |
| | 8 | 1.8 | 3.8 | 0.00 | 0.00 | 3.0 | 69.3 |
| | 9 | 3.0 | 1.2 | 0.00 | 0.00 | 3.6 | 71.3 |
| Y$_2$O$_3$-Nb$_2$O$_5$ | 10 | 3.0 | 0.0 | 0.50 | 0.00 | 10.1 | 83.8 |
| | 11 | 4.2 | 0.0 | 0.20 | 0.00 | 3.9 | 73.3 |
| | 12 | 4.2 | 0.0 | 0.70 | 0.00 | 4.7 | 75.4 |
| | 13 | 4.2 | 0.0 | 0.75 | 0.00 | 4.9 | 74.9 |
| | 14 | 4.2 | 0.0 | 0.80 | 0.00 | 7.5 | 75.9 |
| | 15 | 4.2 | 0.0 | 1.00 | 0.00 | 9.6 | 75.7 |
| | 16 | 4.2 | 0.0 | 1.20 | 0.00 | 9.9 | 76.7 |
| | 17 | 4.2 | 0.0 | 1.40 | 0.00 | 10.7 | 77.6 |
| | 18 | 5.0 | 0.0 | 1.00 | 0.00 | 6.7 | 76.7 |
| | 19 | 5.0 | 0.0 | 1.20 | 0.00 | 9.0 | 77.9 |
| | 20 | 5.0 | 0.0 | 1.40 | 0.00 | 9.7 | 78.2 |
| | 21 | 5.6 | 0.0 | 0.50 | 0.00 | 2.8 | 82.6 |
| | 22 | 5.6 | 0.0 | 1.00 | 0.00 | 3.6 | 86.4 |
| Y$_2$O$_3$-Ta$_2$O$_5$ | 23 | 3.0 | 0.0 | 0.00 | 0.50 | 11.3 | 83.8 |
| | 24 | 4.2 | 0.0 | 0.00 | 0.75 | 7.6 | 75.9 |
| | 25 | 4.2 | 0.0 | 0.00 | 1.00 | 9.7 | 77.1 |
| | 26 | 5.0 | 0.0 | 0.00 | 1.00 | 8.7 | 78.0 |
| | 27 | 5.0 | 0.0 | 0.00 | 1.20 | 9.3 | 79.8 |
| Yb$_2$O$_3$-Nb$_2$O$_5$ | 28 | 0.0 | 4.2 | 0.75 | 0.00 | 5.0 | 77.2 |
| | 29 | 0.0 | 4.2 | 1.00 | 0.00 | 5.2 | 77.3 |
| | 30 | 0.0 | 4.2 | 1.50 | 0.00 | 9.6 | 79.2 |
| Y$_2$O$_3$-Yb$_2$O$_3$-Nb$_2$O$_5$ | 31 | 1.8 | 1.7 | 0.30 | 0.00 | 5.0 | 74.6 |
| | 32 | 1.8 | 1.7 | 0.50 | 0.00 | 6.0 | 75.3 |
| | 33 | 1.8 | 1.7 | 0.75 | 0.00 | 10.4 | 75.5 |
| | 34 | 1.8 | 1.7 | 1.00 | 0.00 | 11.7 | 76.8 |
| | 35 | 1.8 | 1.7 | 1.50 | 0.00 | 12.3 | 78.2 |
| | 36 | 1.8 | 2.4 | 0.75 | 0.00 | 6.0 | 75.8 |
| | 37 | 1.8 | 2.4 | 1.00 | 0.00 | 10.1 | 77.0 |
| | 38 | 1.8 | 2.4 | 1.50 | 0.00 | 11.5 | 78.3 |
| | 39 | 1.8 | 3.8 | 0.50 | 0.00 | 3.5 | 76.2 |
| | 40 | 1.8 | 3.8 | 1.00 | 0.00 | 4.2 | 78.9 |
| | 41 | 3.0 | 1.2 | 1.00 | 0.00 | 8.9 | 76.8 |
| Y$_2$O$_3$-Nb$_2$O$_5$-Ta$_2$O$_5$ | 42 | 4.2 | 0.0 | 0.30 | 0.30 | 5.7 | 77.7 |
| | 43 | 4.2 | 0.0 | 0.50 | 0.50 | 11.3 | 77.5 |
| | 44 | 4.2 | 0.0 | 0.80 | 0.80 | 9.0 | 77.7 |

The crushed powders *A* or *B* were placed into a metal die (20 mm in diameter) and pressed to a disk under 0.78 MPa, subsequently, the disks were treated by CIP (Cold Isostatic Pressing) under 196 MPa. The pressed disks were finally sintered at 1500 °C for 2 h. The final dimensions of the sintered disks were 14.5 ± 0.1 mm in diameter and 1.8 ± 0.3 mm in thickness. The thickness of the disk was adjusted to 1.5 ± 0.01 mm by grinding and mirror polishing. Five specimens were prepared for each composition. The polished sintered disks were used to measure fracture toughness and opacity. The disks were characterized by X-ray diffractometry.

## 2.2. Fracture Toughness

There are various determination methods for the fracture toughness of zirconia. It is often reported that the fracture toughness values are different depending on the determination method and calculated formula [23–25]. In the present study, the Indentation

fracture (IF) method was employed because of easy specimen preparation and data reproducibility. According to the IF method specified in JIS R 1607: 2015 "Testing methods for fracture toughness of fine ceramics at room temperature", fracture toughness values were determined using a Vickers hardness tester (MV-1, Matsuzawa Co., Ltd., Akita, Japan) for each sintered and polished disk according to the following equation:

$$K_{1c} = 0.18\ (E/H_v)^{0.5}(P/c^{1.5}), \tag{1}$$

where $K_{1c}$ is the fracture toughness (Pa m$^{\frac{1}{2}}$), $E$ is the modulus of elasticity (Pa) (206 GPa was employed in this study), $H_v$ is Vickers hardness (Pa) [$H_v = 0.1891P/(2a)^2$], $P$ is the indentation load (N) (98 N was employed in this study), $c$ is the half of the average crack length (m), and $a$ is the half of the average diagonal length of the indenter (m). The measurements were repeated 5 times for each composition, and the average value was calculated.

### 2.3. Opacity

Using a spectrophotometer (CM-3700d, Konica Minolta Co., Ltd., Tokyo, Japan), the reflectance ($R_1$ and $R_0$) of a disk-shaped specimen with a thickness of 1.5 mm was measured against the black and white background, respectively. The opacity (%) is automatically calculated from the measurement results according to the formula:

$$\text{Opacity (\%)} = R_1/R_0 \times 100, \tag{2}$$

when measuring the reflectance from 360 to 740 nm, a calibration plate (CM-A90) and a zero-calibration box (CM-A94) was used as a white and black background, respectively. Measurement geometry was set to the SCI reflectance, 8 mm measurement diameter, 100% Full UV condition, 10° viewing angle, and D65 light source. The measurements were repeated 3 times for each composition, and the average value was calculated.

### 2.4. Statistical Analysis

Multiple regression analysis was performed using Microsoft Office 365 EXCEL. The stepwise forward selection procedure was used to select the best-fitting regression model of the form:

$$Y = \beta_0 + \beta_1 X_1 + \beta_2 X_2 + \beta_3 X_3 + \cdots + \beta_k X_k, \tag{3}$$

where $Y$ is a dependent valuable, $\beta_k$ stands for the regression coefficients, and $X_k$ is the selected independent variable. In the present study, the dependent variables were the toughness and the opacity, and the independent variables were the contents of $Y_2O_3$, $Yb_2O_3$, $Nb_2O_5$, and $Ta_2O_5$.

### 2.5. X-Ray Diffractometry

X-ray diffraction (XRD) patterns of the polished specimen were measured by an X-ray diffractometer (Ultima IV, Rigaku Corporation, Tokyo, Japan) in which scans were conducted at 40 kV and 40 mA between 72 and 76 in $2\theta$ at 0.4°/min using Cu K$\alpha$ radiation. The XRD of 26 specimens (No. 2–20, 22–27 and 30) in which tetragonal and/or cubic crystals exist in a simple composition were measured. Using the analysis software (PDXL2), the lattice constants of the tetragonal $c$-axis and $a$-axis were derived from (004) and (400) peak positions around 73° and 74.5°, respectively. The tetragonal c/a axis lattice constant ratios were derived from these values.

### 2.6. Translucency Parameter

Two kinds of specimens (No. 3 and 16) having different thicknesses (0.28–1.27 mm) were tested for degree of translucency using a portable colorimeter (CR-200, Konica Minolta Co., Ltd., Tokyo, Japan). Three measurements were taken for each specimen on white and black backgrounds and the average of each parameter ($L^*$, $a^*$ and $b^*$) was recorded.

The values were used to calculate the translucency parameter (*TP*) according to the following formula:

$$TP = [(L^*_b - L^*_w)^2 + (a^*_b - a^*_w)^2 + (b^*_b - b^*_w)^2]^{1/2}, \qquad (4)$$

where the subscript *b* is for black and *w* for white.

## 3. Results

### 3.1. Fracture Toughness and Opacity

The experimental sintered zirconia disks contain 0.0–5.6 mol% of $Y_2O_3$, 0.0–4.2 mol% of $Yb_2O_3$, 0.00–1.50 mol% $Nb_2O_5$, and/or 0.00–1.20 mol% $Ta_2O_5$ as stabilizers. The fracture toughness and the opacity varied with the composition as shown in Table 1.

Figure 1 shows the effect of $Y_2O_3$ content on (a) fracture toughness and (b) opacity of $ZrO_2$-$Y_2O_3$. Both properties decreased significantly with increasing $Y_2O_3$ content. Figure 2 shows the $Yb_2O_3$ content dependence on (a) fracture toughness and (b) opacity of $ZrO_2$-$Y_2O_3$-$Yb_2O_3$. Both properties decreased with $Yb_2O_3$ content. Opacity varied significantly with $Yb_2O_3$ content, while toughness did not change much, regardless of $Y_2O_3$ content.

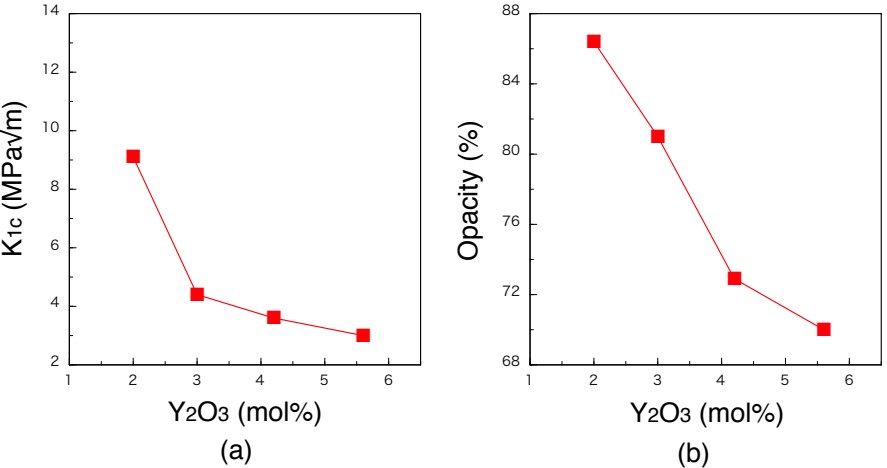

**Figure 1.** The effect of $Y_2O_3$ content on (**a**) fracture toughness and (**b**) opacity of $ZrO_2$-$Y_2O_3$ (No. 1–4).

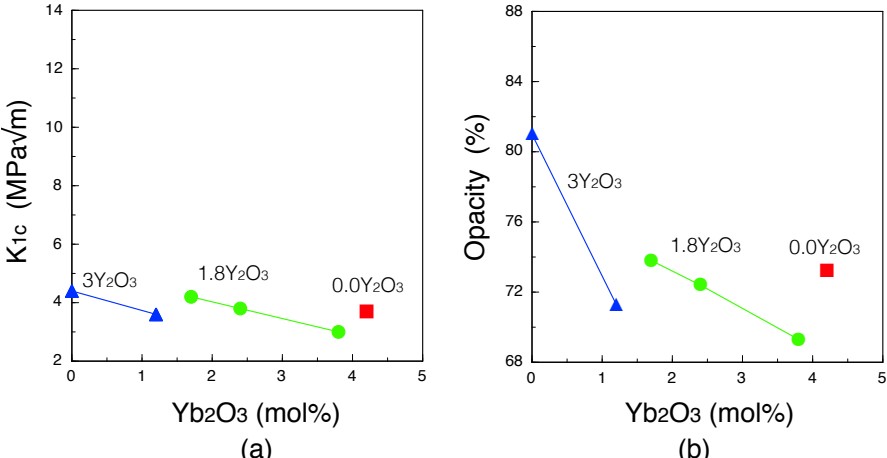

**Figure 2.** The effect of $Yb_2O_3$ content on (**a**) fracture toughness and (**b**) opacity of $ZrO_2$-$Y_2O_3$-$Yb_2O_3$ (No. 2 and 5–9).

Figure 3 shows the effect of $Nb_2O_5$ content on (a) fracture toughness and (b) opacity of $ZrO_2$-$Y_2O_3$-$Nb_2O_5$ and $ZrO_2$-$Yb_2O_3$-$Nb_2O_5$. Both properties increased with $Nb_2O_5$ content. The change differs depending on the $Y_2O_3$ content. At 4.2 mol% $Y_2O_3$, the fracture

toughness increased rapidly with $Nb_2O_5$ above 0.8 mol%, but the opacity did not change much; at 5.6 mol% $Y_2O_3$, the change in fracture toughness due to $Nb_2O_5$ was small, but the opacity increased. It is unclear whether the effect of $Nb_2O_5$ changes depending on whether the co-doping material is $Y_2O_3$ or $Yb_2O_3$.

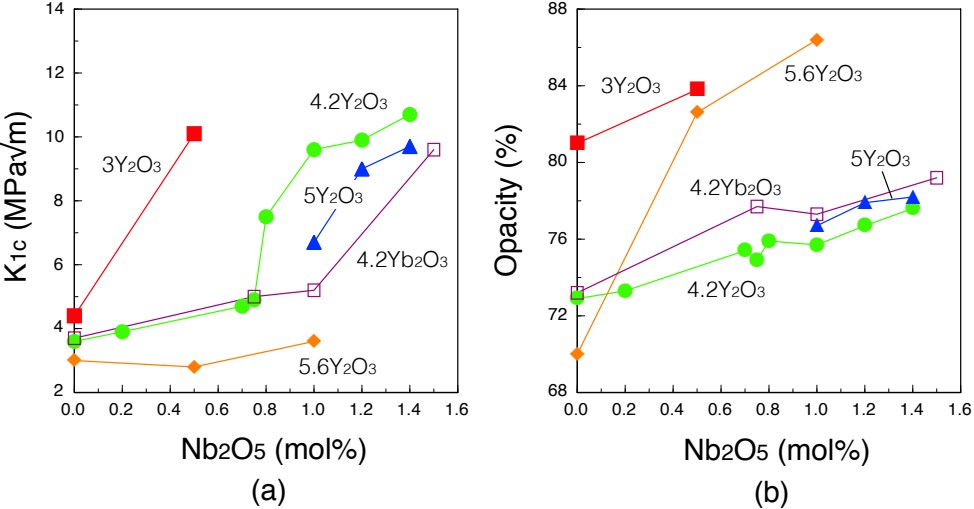

**Figure 3.** The effect of $Nb_2O_5$ content on (**a**) fracture toughness and (**b**) opacity of $ZrO_2$-$Y_2O_3$-$Nb_2O_5$ (No. 2–5 and 10–22) and $ZrO_2$-$Yb_2O_3$-$Nb_2O_5$ (No. 28–30).

Figure 4 shows the effect of $Nb_2O_5$ content on (a) fracture toughness and (b) opacity of $ZrO_2$-$Y_2O_3$-$Yb_2O_3$-$Nb_2O_5$. Both properties increased with $Nb_2O_5$ content. When both $Y_2O_3$ and $Yb_2O_3$ are contained, the effect of the combined content of both $Y_2O_3$ and $Yb_2O_3$ on the toughness and opacity of $Nb_2O_5$ is equivalent to that of each alone.

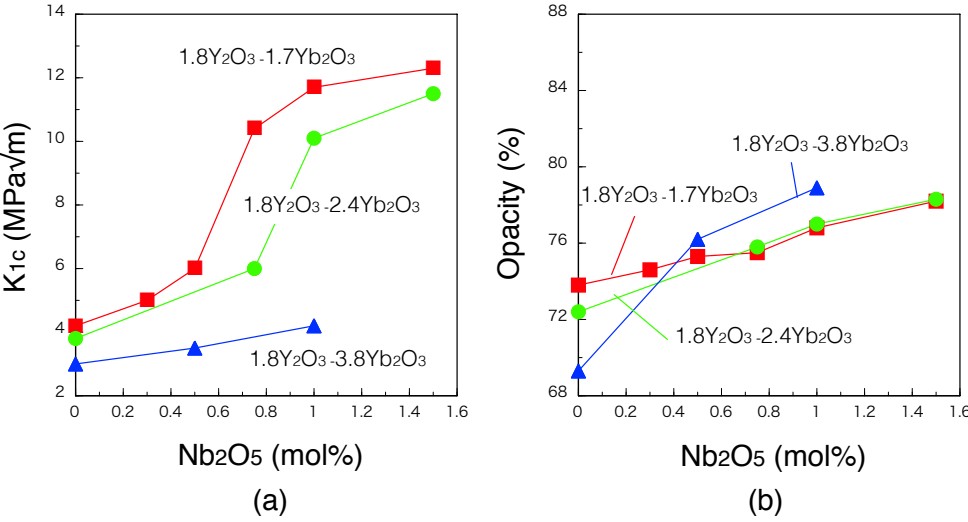

**Figure 4.** The effect of $Nb_2O_5$ content on (**a**) fracture toughness and (**b**) opacity of $ZrO_2$-$Y_2O_3$-$Yb_2O_3$-$Nb_2O_5$ (No. 6–8 and 31–40).

Figure 5 shows the effect of $Ta_2O_5$ content on (a) fracture toughness and (b) opacity of $ZrO_2$-$Y_2O_3$-$Ta_2O_5$. Both properties increased with $Ta_2O_5$ content. Toughness varied significantly with $Ta_2O_5$ content, while opacity did not change much, regardless of the $Y_2O_3$ content.

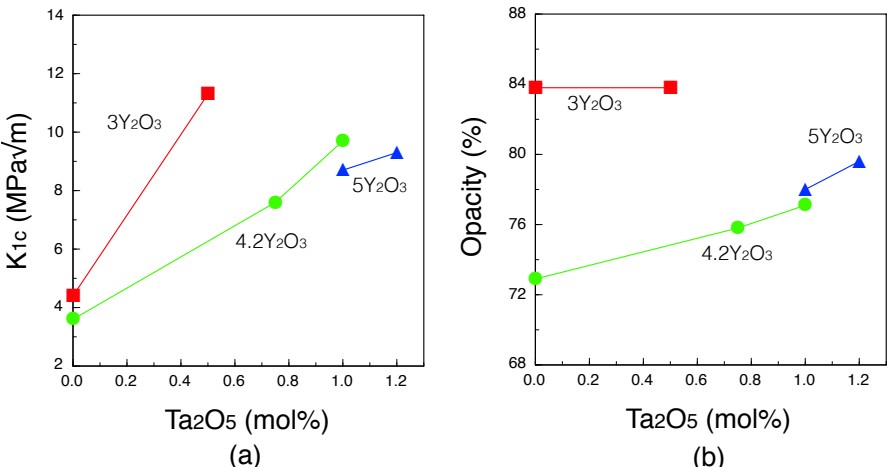

**Figure 5.** The effect of $Ta_2O_5$ content on (**a**) fracture toughness and (**b**) opacity of $ZrO_2$-$Y_2O_3$-$Ta_2O_5$ (No. 2, 3 and 23–27).

Figure 6 shows the effect of the total $Nb_2O_5$ and $Ta_2O_5$ content on (a) fracture toughness and (b) opacity of $ZrO_2$-$Y_2O_3$-$Nb_2O_5$-$Ta_2O_5$. Both properties increased with increasing total $Nb_2O_5$ and $Ta_2O_5$ content; the fracture toughness reached its maximum value when the total $Nb_2O_5$ and $Ta_2O_5$ content was 1 mol%, but the change in opacity was small.

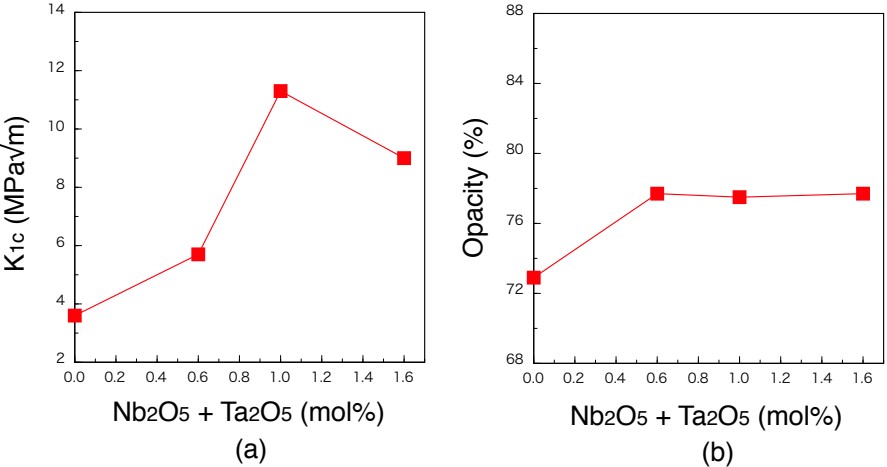

**Figure 6.** The effect of the total $Nb_2O_5$ and $Ta_2O_5$ content on (**a**) fracture toughness and (**b**) opacity of $ZrO_2$-$Y_2O_3$-$Nb_2O_5$-$Ta_2O_5$ (No. 2, 3 and 23–27).

### 3.2. Multiple Regression Analysis

Multiple regression analysis was performed to comprehensively evaluate the results of 44 types of combinations of the above various compositions. Multiple regression analysis revealed the following equations:

$$\text{Toughness (MPa}\sqrt{m}) = 12.314 - 2.008C_Y - 2.177C_{Yb} + 4.620C_{Nb} + 5.899C_{Ta}, \tag{5}$$

$$\text{Opacity (\%)} = 81.184 - 1.327C_Y - 1.819C_{Yb} + 2.787C_{Nb} + 3.047C_{Ta}, \tag{6}$$

where $C_Y$, $C_{Yb}$, $C_{Nb}$, and $C_{Ta}$ are the contents of $Y_2O_3$, $Yb_2O_3$, $Nb_2O_5$, and $Ta_2O_5$ (mol%), respectively. The multiple regression coefficients for the toughness and the opacity were 0.883 and 0.497, respectively. The multiple regression coefficients for the toughness were relatively high and much higher than that for opacity. Depending on the sign of the coefficient for each content, it was shown that the fracture toughness and the opacity decreased with $Y_2O_3$ and $Yb_2O_3$ contents but increased with $Nb_2O_5$ and $Ta_2O_5$ contents.

This comprehensive evaluation can be confirmed in the above graphs (Figures 1–6) as a function of each stabilizer's content.

### 3.3. Microstructural Change

Figure 7 shows X-ray diffraction (XRD) patterns of (a) $ZrO_2$-$Y_2O_3$-$Yb_2O_3$, (b) $ZrO_2$-$4.2Y_2O_3$-$Nb_2O_5$, and (c) $ZrO_2$-$Y_2O_3$-$Ta_2O_5$. In the zirconia disks of $ZrO_2$-$Y_2O_3$-$Yb_2O_3$ (Figure 7a), (400) diffraction peak due to cubic phase increased with $Y_2O_3$ and $Yb_2O_3$ contents, and both (004) and (400) peaks due to the tetragonal phase decreased with them. In specimens of No. 4 and No. 8, which contain large amounts of Y and Yb, only the cubic diffraction peak was observed, and tetragonal diffraction peaks were not observed. On the other hand, in a specimen of No. 2, which has a low Y content, only tetragonal diffraction peaks were observed, and cubic diffraction peak was not observed. There was no significant difference in the effect of $Y_2O_3$ and $Yb_2O_3$ content on the diffraction pattern. In the zirconia disks of $ZrO_2$-$4.2Y_2O_3$-$Nb_2O_5$ (Figure 7b), (400) diffraction peak due to the cubic phase changed little with $Nb_2O_5$ content, whereas both (004) and (400) peaks due to the tetragonal phase remarkably increased with it. In the zirconia disks of $ZrO_2$-$Y_2O_3$-$Ta_2O_5$ (Figure 7c), (400) diffraction peak due to cubic phase increased with $Y_2O_3$ content but not $Ta_2O_5$ content. Both (004) and (400) peaks due to the tetragonal phase decreased with $Y_2O_3$ content but did not change with $Ta_2O_5$ content. There was a slight difference in the effect of $Nb_2O_5$ and $Ta_2O_5$ content on the diffraction pattern. The effect of $Nb_2O_5$ was slightly larger than that of $Ta_2O_5$. As a result, the tetragonal diffraction peaks were observed in 24 out of 26 specimens, and the cubic diffraction peak was observed in 19 specimens.

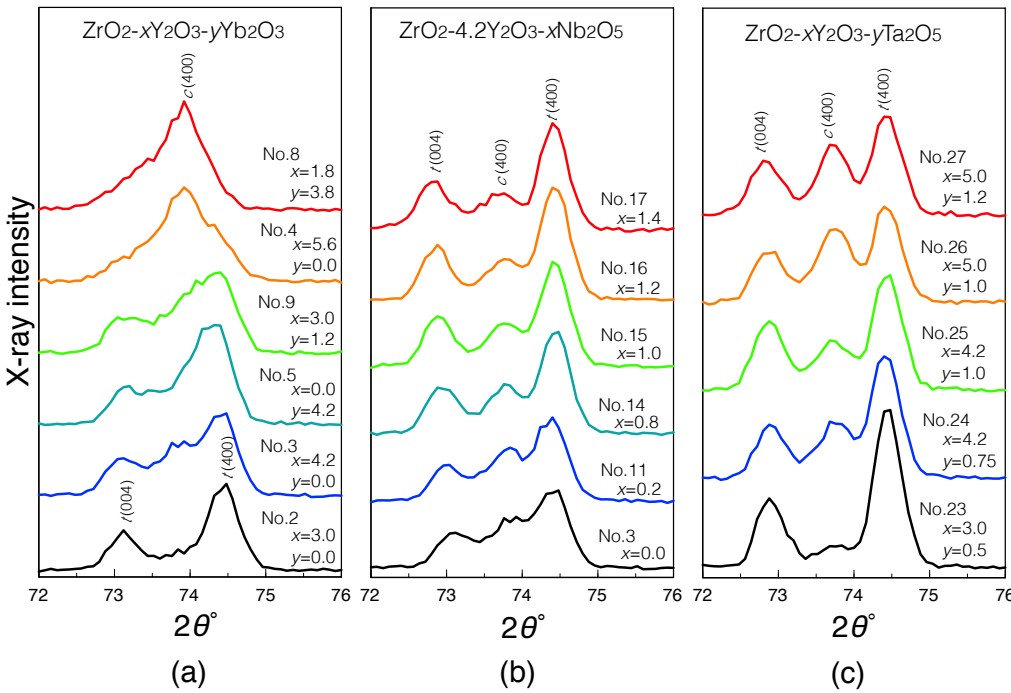

**Figure 7.** XRD patterns of (**a**) $ZrO_2$-$xY_2O_3$-$yYb_2O_3$ ($x$= 0–5.6 and $y$ = 0–3.8 mol%), (**b**) $ZrO_2$-$4.2Y_2O_3$-$xNb_2O_5$ ($x$ = 0–1.4 mol%), and (**c**) $ZrO_2$-$xY_2O_3$-$yTa_2O_5$ ($x$ = 3.0–5.0 and $y$ = 0.5–1.2 mol%).

Figure 8 shows the effect of the c/a axis ratio of the tetragonal $ZrO_2$ phase on (a) the fracture toughness value and (b) the opacity of $ZrO_2$-$Y_2O_3$, $ZrO_2$-$Yb_2O_3$, $ZrO_2$-$Y_2O_3$-$Yb_2O_3$, $ZrO_2$-$Y_2O_3$-$Nb_2O_5$, $ZrO_2$-$Yb_2O_3$-$Nb_2O_5$, and $ZrO_2$-$Y_2O_3$-$Ta_2O_5$. The correlation factors for $K_{1c}$ and opacity to the tetragonal c/a axis ratio were 0.677 and 0.486, respectively. These values denote moderate correlation. So, it suggests that both properties increase with the tetragonal c/a axis lattice constant ratio.

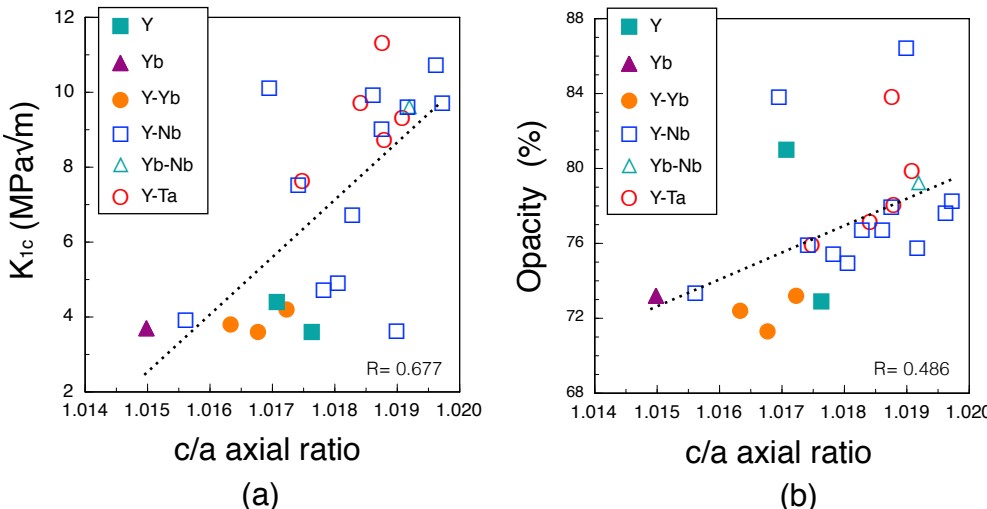

**Figure 8.** (**a**) fracture toughness value and (**b**) opacity of $ZrO_2$-$Y_2O_3$ (Y: No. 2 and 3), $ZrO_2$-$Yb_2O_3$ (Yb: No. 5), $ZrO_2$-$Y_2O_3$-$Yb_2O_3$ (Y-Yb: No. 6, 7 and 9), $ZrO_2$-$Y_2O_3$-$Nb_2O_5$ (Y-Nb: No. 10–20 and 22), $ZrO_2$-$Yb_2O_3$-$Nb_2O_5$ (Yb-Nb: No. 30), and $ZrO_2$-$Y_2O_3$-$Ta_2O_5$ (Y-Ta: No. 23–27) as a function of *c/a* axial ratio of the tetragonal phase.

It is known that the tetragonal *c/a* axis lattice constant ratio, namely tetragonality, means the stability and the crystallinity of the tetragonal $ZrO_2$ phase. When this ratio is 1, the crystal phase is cubic. This indicates that both the fracture toughness and the opacity increased with the tetragonality. Especially, the fracture toughness to the tetragonal *c/a* axis lattice constant ratio has a higher correlation than the opacity to it. Considering the cubic phase change in XRD patterns in Figure 7, it is judged that the opacity is more dependent on the cubic lattice size than on the tetragonal *c/a* axis ratio.

Figure 9 shows the photograph in transmission light of $ZrO_2$-4.2 mol% $Y_2O_3$ with and without 1.2 mol% $Nb_2O_5$ (No. 16 and 3) having different thicknesses. It can be confirmed that a thinner plate has higher translucency and the addition of $Nb_2O_5$ decreased the translucency. Figure 10 shows the thickness effect on the translucency parameter of these specimens. The translucency parameter increased with the decrease in thickness. The translucency parameter of 4.2 mol% $Y_2O_3$-1.2 mol% $Nb_2O_5$ (No. 16) is 38% smaller than that of 4.2 mol% $Y_2O_3$ (No. 3). However, the fracture toughness of the former is 2.75 times higher than that of the latter as shown in Table 1. The increase in fracture toughness is much greater than the decrease in translucency.

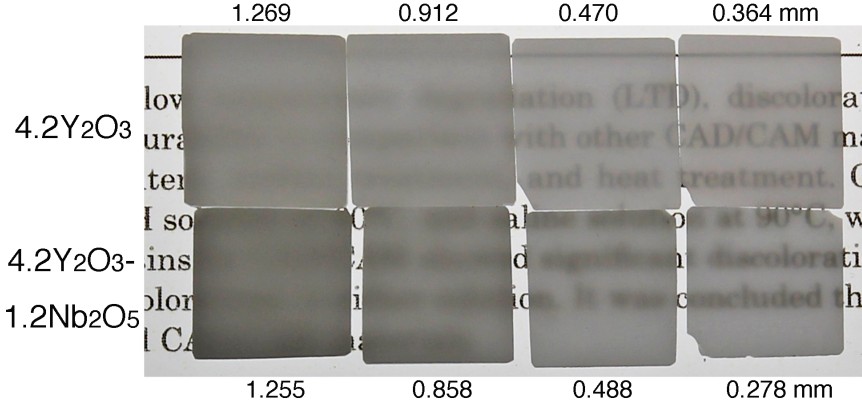

**Figure 9.** Photograph in transmission light of $ZrO_2$-4.2 mol% $Y_2O_3$ with and without 1.2 mol% $Nb_2O_5$ (No. 16 and 3) having 0.278–1.255 and 0.364–1.269 mm in thickness, respectively.

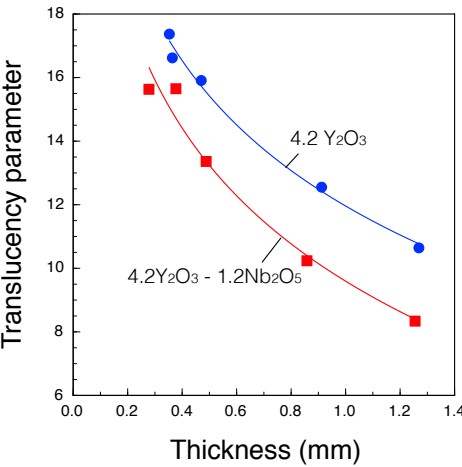

**Figure 10.** Thickness effect on translucency parameter of $ZrO_2$—4.2 mol% $Y_2O_3$ with and without 1.2 mol% $Nb_2O_5$ (No. 16 and 3).

## 4. Discussion

The translucency of PSZ is more complicated than TZP. The grain size is not so effective on PSZ. We demonstrated that the translucency did not change with the final firing temperature over 1450 °C, but the grain size remarkably increased with it [3]. It seems that the translucency of PSZ is independent of the grain size. The translucency of PSZ strongly depends on the content of the cubic phase and increases with it. Since the tetragonal phase is optical anisotropy to reduce translucency due to the birefringence of polycrystals and the cubic phase is optical isotropy to produce high translucency. It is known that the fracture toughness of PSZ depends on the amount of tetragonal phase. This study revealed that the fracture toughness of PSZ also depends on the crystallinity of the tetragonal phase as shown in Figure 8. The opacity, namely the inverse of translucency, has low dependence on it. The crystallinity inevitably changed with the addition of elements as stabilizers. It is confirmed that the addition of trivalent cations Y and Yb reduced the fracture toughness and the opacity. In contrast, the addition of pentavalent cations Nb and Ta increased both properties. No clear difference was found in the effects of Y and Yb, and the effects of Nb and Ta.

Table 2 shows the valence and the ionic radii of co-doped elements employed in this study [26]. Y and Yb have larger ionic sizes and lower charge states than Zr. Nb and Ta have smaller ionic sizes and higher charge states than Zr.

**Table 2.** Valence and ionic radii of co-doped elements used in this study.

| Group | Atomic No. | Element | Valence | Coordination and Ionic Radii (pm) | | |
|---|---|---|---|---|---|---|
| | | | | VI | VII | VIII |
| Trivalent | 39 | Y | 3+ | 90 | 96 | 101.9 |
| | 70 | Yb | 3+ | 86.8 | 92.5 | 98.5 |
| Tetravalent | 40 | Zr | 4+ | 72 | 78 | 84 |
| Pentavalent | 41 | Nb | 5+ | 64 | 69 | 74 |
| | 73 | Ta | 5+ | 64 | 69 | 74 |

When trivalent Y and/or Yb substitute for Zr atoms, the oxygen vacancies introduced by these oversized cations are located as nearest neighbors to Zr atoms for charge compensation, leaving 8-fold oxygen coordination [27]. Then, these cations stabilize the high-temperature polymorphs (cubic and tetragonal).

On the other hand, when pentavalent Nb and/or Ta are co-doped with Y and/or Yb, these cations allow the charge compensating pair such as Y-Nb, Yb-Nb, Y-Ta, and Yb-Ta to

stabilize the tetragonal structure and increase tetragonality due to the removing oxygen vacancies [13,19,22,28]. Kim et al. indicated that the diminution of the oxygen vacancies is responsible for the increase in fracture toughness [13]. Furthermore, this effect due to the pentavalent cations depends on the amounts of trivalent cations stabilized to $ZrO_2$ [19–21]. Guo reported that up to 1.5 mol% $Nb_2O_5$ did not change the cubic structure of 9 mol% $Y_2O_3$ stabilized $ZrO_2$ [29]. The present study also indicated similar results shown in Figure 7. Opacity seems to depend strongly on the amount of the cubic phase and moderately on the cubic lattice size, independent of both the tetragonal and the cubic crystallinity.

Trivalent cations have the effect of lowering the opacity, i.e., increasing the translucency, but decreasing fracture toughness. Whereas pentavalent cations have a greater effect on increasing fracture toughness, the change in opacity is small. This is because pentavalent cations are independent of the formation of cubic crystals but contribute to the stability of tetragonal crystals. On the other hand, dental restorative materials require not only high mechanical strength but also high translucency, i.e., low opacity. Considering the manufacturing process, a composition as simple as possible is desirable. Furthermore, raw materials having low unit prices and ease of availability are also preferable. Although other elements may have similar effects, they may change the color tone such as Ce, and their biological stability is mostly unconfirmed. Therefore, the elements that can be added to zirconia are limited. The elements employed in this study have been already used in dental alloys and ceramics. Therefore, it is practical to employ the addition of Y co-doped with Nb and the following composition is proposed as the optimum composition. A total of 3–4.2 mol% $Y_2O_3$ and/or $Yb_2O_3$ stabilized zirconia with up to 1.5 mol% $Nb_2O_5$ has a high fracture toughness value (4.5 MPa$\sqrt{}$m or more) and low opacity (at a thickness of 1.5 mm of 80.0% or less). Although the translucency slightly decreased with the co-doping with pentavalent cations, the fracture toughness significantly increased with them.

Bending strength was employed to evaluate as representative mechanical strength for dental materials. However, it is reported that the fracture strengths for clinical design strongly depend on the fracture toughness of the materials [30]. We also reported that the bending strength of zirconia is proportional in relation to its fracture toughness [3]. Therefore, fracture toughness was employed to evaluate the adaptability of the zirconia to dental restoratives in this study. Then, it can be expected that the novel zirconia having these properties is suitable for dental restorations. Even if the wall thickness is reduced to improve the translucency of the restoratives, the fracture strength can be maintained sufficiently high due to its high fracture toughness. Then, the novel zirconia can reduce the amount of tooth substance to be removed and subsequently contribute to the realization of gentle treatment according to minimum intervention. In other words, since this new zirconia has sufficiently high fracture toughness, even a thin-walled dental restoration can stably maintain its function in the oral cavity.

The limit of the present study is that the suitability of new zirconia for dental prostheses was evaluated only by fracture toughness and opacity. Further investigations are needed to evaluate the other properties such as bending strength, low-temperature degradation, and operability for dental technicians to pragmatic judge clinical performance.

## 5. Conclusions

Forty-four kinds of zirconia were prepared by combining yttrium (Y), ytterbium (Yb), niobium (Nb), and tantalum (Ta) oxide as stabilizers, and were determined for fracture toughness and opacity. The results revealed that adding of the trivalent cations Y and/or Yb reduced fracture toughness and opacity, whereas the addition of pentavalent cations Nb and/or Ta to $ZrO_2$ stabilized with trivalent cations increased both properties. There was no clear difference in the effects of Y and Yb, and Nb and Ta. Considering many factors, the following composition is optimal: 3–4.2 mol% $Y_2O_3$ and/or $Yb_2O_3$ stabilized zirconia with up to 1.5 mol% $Nb_2O_5$ has sufficiently high fracture toughness values and sufficiently high translucency suitable for dental restorations.

**Author Contributions:** Conceptualization, S.B. and Y.Y.; Data curation, Y.Y., T.S. and Y.M.; Formal analysis, S.B.; Investigation, Y.Y. and Tsutomu Sugiyama; Visualization, S.B.; Writing—original draft, S.B. and Y.Y.; Writing—review and editing, S.B. and Y.Y. All authors have read and agreed to the published version of the manuscript.

**Funding:** This research received no external funding.

**Institutional Review Board Statement:** Not applicable.

**Informed Consent Statement:** Not applicable.

**Data Availability Statement:** Not applicable.

**Acknowledgments:** The authors would like to thank Hiroshi Tenjikukatsura, KCM Corporation, for allowing us to use the preparation equipment and the measurement devices.

**Conflicts of Interest:** The authors declare no conflict of interest.

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
