# Peer review of "Translucent and Highly Toughened Zirconia Suitable for Dental Restorations"

_prosthesis, doi:10.3390/prosthesis5010005_

Round 1
Reviewer 1 Report
Dear Author, your paper is dealing with a much debated topic. This reviewer found it very interesting,nevertheless fond some puzzling points.
1 - why PSZ? The standard zirconia in dentistry is 3Y-TZP. PSZ is a more complex material, due to its peculiar sintering cycle designed to obtain on cooling the precipitation of fine lenticular tetragonal zirconia within large (40-80 micron) cubic grains.
2 - The discussion of the results obtained is extensively dealing with mechanical properties (Toughness). This reviewer recommends to insert data on bending strength. This is mandatory, because to make the mechanical comparison of materials on toughness only may lead to misleading conclusions.
3 - A key point is missing in the discussion of your interesting results. It is known that light trasmittance is depending on the birifrangence of the microstructure. A key parameter is the grain size and its distribution. This data is missing from your manuscript. This reviewer recommends: a) to measure the grain size distribution in the materials tested; b) to discuss the measures of light trasmittance with respect to grain size.
3 - This reviewer recommends to discuss your results in the light of the following papers:
- Shahmiri R, et al. J Prost Dent 2018; 119(1):36-46.
- Stawarczyk B, et al. Dent Mater 2014;33(5):591-8.
Reviewer 2 Report
Dear authors,
please have a look in the comments below before proceeding with your manuscript:
-Are you sure the abstract structure is according to the journal?
-"rare earth oxide": better "oxides of rare earths"
-"is dissolved": if sol-gel process is present ok, if the synthesis is unknown maybe "is contained" is more general.
-"yttria oxide": incorrect, either yttrium oxide or yttria
-"of the sintered body": what do you mean? rephrase maybe
-"co-doped elements": incorrect, "co-doped zirconia with elements" better
-"were prepared as stabilizers by": "were prepared with stabilizers got by" better
-line 71: "in solid form" instead of "sol"
-line 73: never use a comma (,) before "and" or "or" when simple parathesis of similar things is the case. Check and correct throughout the text where needed.
-"h" instead of hours, in SI units. Check and correct throughout the text where needed.
-1500oC: always keep a space between the numbers and the units, 1500 oC. Check and correct throughout the text where needed.
-"by grinding and mirror polishing": method or tool?
-Table 1: some rows are underlined?
-"b is for black": better "where subscripts b is for black.."
-Figures: the axes' labels need improvement: either the font style or size or bold.. anyway, uniform lebeling
-0 Y2O3 on charts means no yttria?
-for all expressions including "content dependence on (a) fracture toughness and (b) opacity": the content does not depend on the properties! The opposite, the properties of the materials depend upon the compound, the structure or the proportion. Thus you may replace with "content effect on (a) fracture toughness and (b) opacity" or "(a) fracture toughness and (b) opacity dependence on content"
likewise to expressions like "the dependence of the c/a axis ratio of the tetragonal ZrO2 phase"
-line 204: justified alignement for paragraph
- [Figure 7 (a)]: better in parenthesis (Figure 7a), likewise the rest
-specimens No. ..., without of, Y-content with dash
-Table 2: valence 3+, 5+
-"fracture toughness13" 13 is areference?
- i.e. in italics font
-"zirconia were prepared as stabilizers": those powders would be used as stabilizers or as reinforcing agents? Additionaly to fillers?
-I would propose some more comments in Discussion part
Reviewer 3 Report
Highly toughened zirconia ceramics have been developed by adding different elements and their mechanical and optical properties have been evaluated for dental restoration applications. The paper presents detailed investigation on 44 different types of samples. XRD studies were conducted to identify the differences in materials microstructural characteristics. Regression analysis was carried out to identify any relationship between the elemental composition and their mechanical and optical properties. However the following points needs to be addressed before acceptance of the paper.
The abstract should provide the details of the elemental % used in different ceramics.
The axis titles in the figures are very close to the axis. Fix this please
Needs to add further details about the rationale for different combinations of elemental contents in the specimens
Clinical significance needs to be added
Any limitation of the study and future plan
Round 2
Reviewer 2 Report
Please,
-remove all erased corrections, fit additions and refromat the paper properly.
-Table 1: add a column, 1st column, where the groups formed by the intermediate horizontal lines are named or commented
-Improve English where possible
Author Response
Thank you for your careful review.
As follows, we responded as best we could.
-remove all erased corrections, fit additions, and reformat the paper correctly.
Please use correction history in the Microsoft Word command. You can quickly check the manuscript with and without corrections.
-Table 1: add a column, 1st column, where the groups formed by the intermediate horizontal lines are named or commented
A column was added to the beginning of Table 1 to display the group name.
-Improve English where possible
English has been revised as much as possible. We corrected a few things that should have been italicized. Also, we deleted the superfluous expressions. Furthermore, we found some simple mistakes and corrected them.
